# TAS1R3 Regulates GTPase Signaling in Human Skeletal Muscle Cells for Glucose Uptake

**DOI:** 10.3390/ijms27010103

**Published:** 2025-12-22

**Authors:** Joseph M. Hoolachan, Rekha Balakrishnan, Karla E. Merz, Debbie C. Thurmond, Rajakrishnan Veluthakal

**Affiliations:** Department of Molecular & Cellular Endocrinology, Arthur Riggs Diabetes and Metabolism Research Institute of City of Hope, Duarte, CA 91010, USA; jhoolachan@coh.org (J.M.H.); rbalakrishnan@coh.org (R.B.); kmerz@amgen.com (K.E.M.)

**Keywords:** type 2 diabetes, skeletal muscle, Rac1, TAS1R3 signaling, glucose uptake, insulin resistance

## Abstract

Taste receptor type 1 member 3 (TAS1R3) is a class C G protein-coupled receptor (GPCR) traditionally associated with taste perception. While its role in insulin secretion is established, its contribution to skeletal muscle glucose uptake, a process responsible for 70–80% of postprandial glucose disposal, remains unclear. TAS1R3 expression was assessed in skeletal muscle biopsies from non-diabetic and type 2 diabetes (T2D) donors using qPCR and immunoblotting. Functional studies in human LHCN-M2 myotubes involved TAS1R3 inhibition with lactisole or siRNA-mediated knockdown, followed by the measurement of insulin-stimulated glucose uptake using radiolabeled glucose assays. Rac1 activation and phospho-cofilin were analyzed by G-LISA and Western blotting, and Gαq/11 involvement was tested using YM-254890. TAS1R3 mRNA and protein levels were significantly reduced in T2D skeletal muscle. Pharmacological inhibition or the knockdown of TAS1R3 impaired insulin-stimulated glucose uptake in myotubes. TAS1R3 regulates skeletal muscle glucose uptake through a non-canonical insulin signaling pathway involving Rac1 and phospho-cofilin, independent of IRS1-AKT and Gαq/11 signaling. These findings identify TAS1R3 as a key determinant of Rac1-mediated glucose uptake and a potential therapeutic target for improving insulin sensitivity in T2D.

## 1. Introduction

Over 750 million people globally have prediabetes. Without intervention, 50% of individuals with prediabetes will progress to type 2 diabetes (T2D) annually [1]. In 2021, the global age-adjusted prevalence of impaired glucose tolerance (IGT) among adults aged 20–79 years was estimated at 9.1%, corresponding to approximately 464 million individuals. This prevalence is projected to rise to 10.0% (~638 million individuals) by 2045. Similarly, in 2021, an estimated 5.8% of adults (20–79 years) (~298 million people) had impaired fasting glucose (IFG). By 2045, this figure is expected to increase to 6.5%, affecting roughly 414 million individuals [2]. Skeletal muscle insulin resistance is a major pathological feature in T2D [3]. Peripheral insulin resistance is a key driver of prediabetes and its progression into T2D, with skeletal muscle being a major target [3,4,5]. In healthy individuals, skeletal muscle plays a central role in maintaining whole-body glucose homeostasis, accounting for ~70–90% of insulin-stimulated peripheral blood glucose uptake [6,7,8]. However, prolonged skeletal muscle insulin resistance can drive myopathic comorbidities in T2D patients including progressive muscle loss (sarcopenia) [9], exercise intolerance [10,11], skeletal muscle inflammation [12], and impaired muscle repair [13]. Weight management is a key intervention for mitigating insulin resistance and T2D, and the widely used glucagon-like peptide-1 receptor agonists (GLP-1 RAs) are associated with an elevated risk of sarcopenia [14,15,16,17]. Therefore, novel therapeutic strategies that can effectively prevent or reverse skeletal muscle insulin resistance, albeit without any adverse myopathic risks, are needed. 

Although multiple factors are attributed to the development of skeletal muscle insulin resistance including defective glucose and lipid utilization [18], progressive mitochondrial decline [19], aging [20], and oxidative stress [21], a key feature is the impairment of the canonical (PI3K/AKT) [22,23] and non-canonical (GTPase/Actin remodeling) insulin signaling cascades [24]. Normally, both cascades require the stimulation of the insulin receptors; however, the non-canonical insulin signaling pathway is able to stimulate GLUT4-mediated vesicle translocation to the sarcolemma independent of Insulin receptor substrate 1 (IRS1) activation [25]. Indeed, several G-protein-coupled receptor pathways are able to activate the small Rho-family GTPase Rac1 to promote actin filament remodeling to promote GLUT4 translocation [26]. Although this provides an insulin-independent source for GLUT4 translocation, the upstream regulators that coordinate small GTPase activity in skeletal muscle remain incompletely defined.

One potential candidate to coordinate small GTPase activity in skeletal muscle is the sweet taste receptor type 1 member 3 (TAS1R3), a G protein-coupled receptor (GPCR), ubiquitously expressed in various tissue types including the tongue, pancreatic β-cells [27], gastrointestinal tract [28], and musculoskeletal system [29], to name a few. In addition to TAS1R3, other GPCRs such as β_2_-adrenergic receptors play critical roles in skeletal muscle hypertrophy and atrophy, as well as in adaptive responses to anabolic and catabolic conditions, highlighting the broader importance of GPCR signaling in muscle metabolism. Normally, the nutrient-sensing role of TAS1R3 is dependent upon its dimerization, with TAS1R3/TAS1R3 homodimers [30] and TAS1R2/TAS1R3 heterodimers [31] serving as glucose sensors and TAS1R1/TAS1R3 heterodimers serving as an amino acid sensor [32]. Importantly, the downregulation of TAS1R3 levels coincide with aberrant whole-body glucose homeostasis, as observed with the impaired glucose tolerance in global TAS1R3 knockout mice [33]. Although the prior literature has no direct evidence linking TAS1R3 single-nucleotide polymorphisms to T2D and obesity risks [31], studies on gastric tissue [34] and pancreatic-β cells [27] have shown diminished TAS1R3 expression in obese and T2D patients, suggesting that TAS1R3 activity may be altered in obesity/T2D pathogenesis. Given that a number of conserved molecular pathways play a role in both β-cell GSIS and skeletal muscle insulin-stimulated glucose uptake [35,36,37,38,39,40], it still remains to be determined if skeletal muscle TAS1R3 expression is altered during T2D pathogenesis and whether this is attributed to a potential novel role of TAS1R3 in insulin-stimulated glucose uptake. 

In this study, we hypothesize that TAS1R3 is required for insulin-stimulated glucose uptake in skeletal muscle through Rac1-mediated non-canonical signaling. Using T2D and non-diabetic human skeletal muscle biopsies, as well as clonal human LHCN-M2 myotube cell lines, we aim to elucidate the significance of TAS1R3 signaling in skeletal muscle insulin resistance. Rac1, a key regulator of actin cytoskeleton remodeling necessary for GLUT4 translocation, exhibits impaired insulin-stimulated activation in T2D [24]. This dysregulation hinders glucose uptake and contributes to muscle insulin resistance, underscoring the importance of pathways that intersect with TAS1R3-mediated signaling.

## 2. Results

### 2.1. TAS1R3 Transcript and Protein Levels Are Decreased in T2D Human Skeletal Muscle

To investigate the impact of diabetogenic stress on TAS1R3 expression, we assessed TAS1R3 mRNA levels in human LHCN-M2 myotubes exposed for 16 h to glucolipotoxic (GLT) conditions (25 mM glucose and 200 μM palmitic acid). TAS1R3 transcript levels were significantly reduced in human LHCN-M2 myotubes post-GLT exposure (Figure 1A). In parallel, insulin-stimulated glucose uptake was markedly diminished in these cells (Figure 1B), further indicating the onset of insulin resistance. These findings are consistent with previous reports demonstrating that palmitate exposure induces insulin resistance, thereby reducing glucose uptake in L6 skeletal muscle myotubes and in human primary myotubes [41,42]. Next, we assessed TAS1R3 levels in T2D and non-diabetic (ND) human skeletal muscle biopsies (Table 1). Both TAS1R3 mRNA levels and protein abundances were significantly reduced (>50%) in T2D skeletal muscle versus non-diabetic biopsies (Figure 1C,D). Thus, TAS1R3 deficiency may contribute to peripheral glucose uptake dysfunction in T2D skeletal muscle. 

### 2.2. TAS1R3 Inhibition Impedes Insulin-Stimulated Glucose Uptake via a Non-Canonical Insulin Signaling Pathway

To determine the requirement for TAS1R3 in skeletal muscle peripheral glucose uptake, we evaluated the impact of TAS1R3 (1 mM Lactisole; TAS1R3_(*i*)_) or Gαq/11 (10μM YM-25490; Gαq/11_(*i*)_) inhibition against a vehicle control in insulin-stimulated human LHCN-M2 myotubes by measuring 2-deoxyglucose ([^3^H]2-DG) uptake based on initial dose–response studies and their prior use in glucose-stimulated insulin secretion (GSIS) in pancreatic β-cells [27]. Skeletal muscle is a major site for glucose disposal and plays a critical role in systemic glucose homeostasis. Previous studies have shown that skeletal muscle-specific activation of Gq signaling improves glucose uptake and insulin sensitivity, even under metabolic stress [43]. Additionally, CaSR, a Gαq/11-coupled receptor activated by kokumi substances, enhances sweet, salty, and umami perception via TAS1R3 components [44], highlighting Gαq/11 signaling as a key integrator of nutrient-sensing and metabolic regulation across tissues. This overlap suggests that Gαq/11 signaling integrates nutrient-sensing and metabolic regulation across multiple tissues. Therefore, investigating Gαq/11 in skeletal muscle provides insight into whether TAS1R3 couples Gq mechanisms to downstream signaling components for glucose uptake.

TAS1R3 inhibition significantly impeded glucose uptake by ~50% versus the vehicle in insulin-stimulated human LHCN-M2 myotubes (Figure 2A). We also observed that TAS1R3 inhibition reduced glucose uptake under basal conditions (Figure 2A). Activation of the small GTPase Rac1 is required to mobilize GLUT4-laden vesicles to supply the GLUT4 protein at the sarcolemma to facilitate insulin-stimulated glucose uptake via a non-canonical insulin pathway [45]. We tested the requirement for TAS1R3 in Rac1 activation in human LHCN-M2 myotubes by inhibiting TAS1R3 and comparing with Gαq/11 inhibition. While neither inhibitor altered Rac1-GTP levels at basal conditions (Figure 2B), only TAS1R3 inhibition significantly reduced insulin-stimulated Rac1-GTP levels (Figure 2B). We verified that the decrease in insulin-stimulated glucose uptake in myotubes was indeed TAS1R3-specific via TAS1R3 small interfering RNA (siRNA) knockdown (Figure 2C,D). Thus, these results indicate a selective role for TAS1R3 in insulin-stimulated glucose uptake in human LHCN-M2 myotubes.

We next investigated the impact of TAS1R3 inhibition on canonical and non-canonical insulin signaling cascades [22,23,24,46]. Canonical insulin signaling was assessed in LHCN-M2 myotubes, and TAS1R3 inhibition did not impact the phosphorylation (activation) of canonical insulin cascade components IRS1 (Tyr^608^) and AKT (Ser^473^) (Figure 3A,B). Non-canonical insulin signaling downstream of Rac1 involves Cofilin, a protein that oversees actin remodeling to translocate GLUT4-laden vesicles to the plasma membrane. Cofilin phosphorylation leads to the depolymerization of actin filaments, which impedes GLUT4 vesicle translocation in skeletal muscle cells [47,48]. Vehicle-treated insulin-stimulated myotubes showed >50% decrease in p-Cofilin (relative to total cofilin protein); no effect on p-Cofilin was observed with TAS1R3i-treated insulin-stimulated myotubes (Figure 3C). As observed above (Figure 2), Gαq/11 inhibition did not impact the insulin signaling components (Figure 3A–C). Taken together, our results suggest that TAS1R3 functions in non-canonical insulin signaling via Rac1 activation to promote insulin-stimulated glucose uptake in LHCN-M2 myotubes. 

## 3. Discussion

This study provides mechanistic insights into the role of TAS1R3-mediated monomeric GTPase signaling in skeletal muscle insulin-stimulated glucose uptake. For the first time, we showed that T2D significantly reduces TAS1R3 mRNA and protein levels in human skeletal muscle. Furthermore, we demonstrated that both direct pharmacological inhibition and the siRNA knockdown of TAS1R3 contributes to decreased insulin-stimulated glucose uptake in human LHCN-M2 myotubes, consistent with the impaired glucose tolerance and insulin resistance phenotype observed in the TAS1R3 knockout mice [33]. In this study, TAS1R3 function in skeletal muscle glucose uptake was attributed to the non-canonical Rac1-mediated insulin cascade for GLUT4 vesicle translocation to the sarcolemma, which is independent of Gαq/11 signaling. Furthermore, we observed that TAS1R3 inhibition reduced glucose uptake under basal conditions (Figure 2A), akin to Rac1 GTPase inhibitors on soleus muscle under basal conditions [46]. Additionally, the TAS1R3 blockade of basal glucose suggests that TAS1R3 supports basal glucose uptake, potentially through nutrient-sensing mechanisms intrinsic to skeletal muscle, independently of insulin signaling. Collectively, the results highlight a role for TAS1R3 in skeletal muscle insulin-stimulated glucose uptake. 

A key finding of this study was the essential role that TAS1R3 plays as a GPCR target to initiate the non-insulin signaling cascade in human myotubes for insulin-stimulated glucose uptake via small GTPase Rac1 activation, which in turn promotes actin cytoskeletal remodeling via phospho-mediated inactivation of cofilin for GLUT4-vesicle translocation to the sarcolemma. To our knowledge, TAS1R3 signaling does not directly regulate GLUT4 transcription. The pharmacological studies performed in this paper involve the short-term exposure of TAS1R3 antagonists that would only impact the post-translational modification. Therefore, we do not anticipate any changes in GLUT4 expression levels under these conditions. Although we identified that TAS1R3 regulates this independently of Gαq/11 signaling, the precise trimeric GTPase that links TAS1R3 to downstream pathways remains to be elucidated. One promising G-protein candidate for future studies based on the prior literature is Gα14, based on the finding that FR900359, a selective inhibitor of Gαq/11/14, significantly impairs both GLUT4 translocation and glucose uptake in skeletal muscle cells [43], which YM-254890, a Gαq/11-specific inhibitor, failed to replicate. However, with Gα14 being just one example of the numerous potential G-protein candidates, future studies will need to focus on identifying the downstream regulators that directly link TAS1R3 activity with Rac1-GTP-mediated GLUT4-vesicle translocation for insulin-stimulated glucose uptake in skeletal muscle cells in vitro.

These findings align with emerging evidence from TAS1R2 signaling, where glucose stimulation of this GPCR activates the ERK1/2-dependent phosphorylation of PARP1, a major NAD consumer in skeletal muscle [49]. Muscle-specific TAS1R2 knockout in mice suppressed PARP1 activity, elevated NAD levels, enhanced mitochondrial function, and improved endurance, positioning TAS1R2 as a peripheral energy sensor [49]. Together, these studies underscore the broader role of GPCRs like TAS1R2 and TAS1R3 in regulating skeletal muscle metabolism through distinct but complementary pathways involving glucose sensing, cytoskeletal remodeling, and NAD homeostasis. Future work is needed to identify the specific trimeric GTPase(s) downstream of TAS1R3 and to determine whether TAS1R3, like TAS1R2, contributes to mitochondrial health and endurance capacity.

We showed that TAS1R3 mRNA and protein levels were depleted in T2D skeletal muscle, which was similarly observed in GLT-exposed human LHCN-M2 myotubes, which is used as a model for earlier stages of insulin resistance, suggesting that TAS1R3 levels are depleted in insulin-resistant muscle. To the best of our knowledge, this is the first study that has associated in a clinical context the association of TAS1R3 levels with skeletal muscle insulin resistance. However, we acknowledge that our human biopsy data is limited by a small sample pool (*n* = 6 per group) that could be impacted by donor variability based on the available clinical data showing sex, age, and body mass index (BMI) differences. Thus, larger cohorts are needed that consist of non-insulin-resistant, prediabetes (impaired fasting blood glucose and/or impaired insulin tolerance), and T2D patients to support the decline in TAS1R3 with the progression of insulin resistance. 

Outside of glucose homeostasis, TAS1R3 was shown to be involved in various musculoskeletal system processes, including myogenesis, amino acid sensing via mTOR activation to suppress autophagy [50], and osteoclastogenesis for bone remodeling and homeostasis [51]. Importantly, the long-term secondary complications from prolonged T2D include dysfunctions across these aforementioned functions/pathways culminating in musculoskeletal comorbidities that include reduced bone quality, weakened muscle, and impaired muscle repair. Conversely, the risks which current T2D therapeutic interventions such as GLP-1 RAs pose on exacerbating these musculoskeletal comorbidities highlight the necessity of novel T2D therapeutics to mitigate and/or prevent these events. 

One limitation of the current study is the inability to determine the impact of skeletal muscle-specific TAS1R3 inactivation or depletion on whole-body glucose homeostasis. This highlights the need for cell type-specific, inducible TAS1R3 knockout animal models to enable in vivo investigations. Thus, future studies need to [1] evaluate whether novel TAS1R3 agonists or genetic enrichment strategies can reverse skeletal muscle insulin resistance; and [2] develop skeletal muscle-specific, inducible TAS1R3 overexpression models to assess the potential benefits of TAS1R3 enrichment on primary insulin resistance and secondary muscle pathologies associated with T2D.

## 4. Materials and Methods

### 4.1. LHCN-M2 Myoblast Cell Culture

For the physiological relevance of skeletal muscle glucose uptake, the immortalized human LHCN-M2 skeletal myoblast cell line was used (a generous gift from Dr. Melissa Bowerman, Keele University, UK). LHCN-M2 myoblasts were cultured in complete growth medium (4:1 ratio of 5 mM glucose DMEM) (Thermo Fisher, Waltham, MA, USA) and 5 mM glucose M199 medium (Thermo Fisher, Waltham, MA, USA) supplemented with 15% [*v*/*v*] HI-FBS, 1% [*v*/*v*] antibiotic–antimycotic solution, 20 mM pH 7.4 HEPES buffer, 30 ng/mL Zn_2_SO_4_, 1.4 μg/mL vitamin B12 (Sigma-Aldrich, St. Louis, MO, USA), 55 ng/mL dexamethasone (Sigma-Aldrich, St. Louis, MO, USA), 2.5 ng/mL recombinant human hepatocyte growth factor (Sigma-Aldrich, St. Louis, MO, USA), and 10 ng/mL recombinant human basic FGF (Biopioneer, San Diego, CA, USA) on plastic cell culture dishes pre-coated with 1% [*w*/*v*] autoclaved porcine gelatin (Sigma-Aldrich, St. Louis, MO, USA). At >70% confluence, LHCN-M2 myoblasts were cultured in complete differentiation medium composed of DMEM/M199 4:1, 2% [*v*/*v*] heat-inactivated horse serum (HI-HS) (Thermo Fisher, Waltham, MA, USA), 1% [*v*/*v*] anti-biotic/anti-mycotic solution, 20 mM pH 7.4 HEPES buffer, 30 ng/mL Zn_2_SO_4_, and 1.4 μg/mL vitamin B12 for 6 days with fresh media changes every 48 h in 37 °C and a 5% CO_2_ incubator until multi-nucleated myotubes formed. For glucolipotoxicity (GLT)-induced diabetogenic stress, LHCN-M2 myotubes were exposed to 200 μM palmitate and 25 mM glucose for 16 h. 

### 4.2. Human Skeletal Muscle

Cadaveric non-diabetic and T2D skeletal leg muscles were purchased from the National Disease Research Interchange (NDRI, Philadelphia, PA, USA). See Table 1 for donor information. The samples were snap-frozen and kept at −80 °C until mRNA and protein extraction were carried out.

### 4.3. Quantitative PCR

Total RNA isolation from human donor samples and LHCN-M2 myotubes was performed with TriReagent (Millipore Sigma, St. Louis, MO, USA) as previously described [52] and from the LHCN-M2 cell line with a RNeasy Plus Mini Kit (Qiagen, Germantown, MD, USA) according to the manufacturer’s instructions. Gene expression was assessed using two-step reverse transcription (iScript^TM^ cDNA Synthesis Kit, Bio-Rad, Hercules, CA, USA) and qPCR (iQ SYBR^®^ Green Supermix, Bio-Rad, Hercules, CA, USA). The cycle threshold data was converted to change-fold in expression by the “ΔΔCt” method.

### 4.4. Immunoblot Analysis

Whole-protein lysates of cell cultures and primary human skeletal muscle biopsies were lysed with 1% NP-40 lysis buffer, resolved, and underwent immunodetection as previously described [53]. For the detection of the insulin signaling pathway, membranes were incubated with primary antibodies for IRS (Tyr^608^, 1:1000, Millipore-Sigma, St. Louis, MO, USA), AKT (Ser^473^, 1:1000, Cell Signaling Technology, Danvers, MA, USA) [53], and cofilin (Ser^3^, 1:1000, Santa Cruz Biotechnology, Inc., Dallas, TX, USA) [47]. As a control for equivalent protein-loading, IRS (1:1000, Cell Signaling Technology, Danvers, MA, USA), AKT (1:1000, Cell Signaling Technology, Danvers, MA, USA), and Cofillin (1:1000, Cell Signaling Technology, Danvers, MA, USA) and GAPDH (1:50,000, Invitrogen, Waltham, MA, USA) antibodies were used. 

### 4.5. LHCN-M2 2-Deoxyglucose Uptake 

For the pharmacological analysis, LHCN-M2 myotubes were subjected to two washes with an oxygenated FCB buffer devoid of serum and glucose. This buffer was composed of 125 mM NaCl, 5 mM KCl, 1.8 mM CaCl_2_, 2.6 mM MgSO_4_, 25 mM HEPES, 2 mM pyruvate, and supplemented with 2% (wt/vol) bovine serum albumin (BSA). Subsequently, the myotubes were treated with either vehicle (0.1% *v*/*v* DMSO) or 1 mM lactisole to specifically inhibit TAS1R3, or 10 µM YM-25490, a selective antagonist for Gαq/11 signaling. This treatment was conducted in the aforementioned FCB buffer for 1 h. Following drug exposure, insulin stimulation was performed using 100 nM insulin for 20 min. The myotubes were then assessed for 2-deoxyglucose (2-DG) uptake, following methodologies previously established for L6.GLUT4myc myotubes [40,53,54]. 

### 4.6. Small Interfering RNA Transfection

For TAS1R3 knockdown studies, LHCN-M2 myotubes at differentiation day (D) 4 were transfected with a 50 nM TAS1R3-targeting siRNA (siTAS1R3) sequence or non-specific targeting siRNA control in a drop-wise fashion using RNAiMax and Opti-MEM overnight, as previously described [54]. These myotubes underwent a total of 48 h of transfection prior to the 2-DG uptake protocol (see above).

### 4.7. G-LISA for Rac1 Activation Assay

Rac1 activation was measured in the cell lysate from LHCN-M2 myotubes using the commercially available G-LISA Rac1 Activation Assay from Cytoskeleton Inc. (Denver, CO, USA). In brief, the LHCN-M2 myotubes were washed twice with PBS and then serum- and glucose-starved in oxygenated FCB buffer for one hour. During this time, the myotubes were treated with either vehicle or 1 mM lactisole, or 10 µM YM-254890 to induce the pharmacological inhibition of TAS1R3 (lactisole) or Gq/11 (YM-254890). This was followed by a 5 min stimulation with 100 nM insulin, and comparisons were made to basal controls with and without drug treatment. Rac1 activation was measured in the supernatant using the commercially available G-LISA Rac1 Activation Assay from Cytoskeleton Inc. (Cat # BK 128), as described previously [55]. The intensity of the color formed from the chromogenic substrate was quantified by colorimetry using a BioTek Synergy HTX Multi-Mode Plate Reader (BioTek Instruments Inc., Winooski, VT, USA).

### 4.8. Insulin Signaling Analysis

For the detection of the insulin signaling pathway, LHCN-M2 myotubes underwent 1h of incubation in serum- and glucose-starved oxygenated FCB buffer with either vehicle, 1 mM lactisole, or 10 μM YM-254890 prior to 100 nM insulin stimulation for 10 min, as described for L6.GLUT4myc myotubes [53]. 

### 4.9. Statistics

Data are presented as mean ± SEM and *n* values are indicated in the figures. Differences between two groups were assessed using Student’s *t*-test. Statistically significant differences among multiple groups were evaluated using a one-way or two-way ANOVA followed by the Bonferroni post hoc test. The threshold for statistical significance was set at *p* < 0.05 and all the data were analyzed using the GraphPad Prism software, version 8.3.0. Statistical significance is indicated in the figure legends.

## 5. Conclusions

In conclusion, our findings provide novel insights into the functional activity of TAS1R3 as a modulator of insulin-stimulated glucose uptake via monomeric GTPase Rac1 (Figure 4), independent of Gαq/11 signaling. Furthermore, we demonstrated that TAS1R3 is dysregulated in T2D skeletal muscle, suggesting a role in glucose homeostasis. Nevertheless, these emerging mechanistic insights establish TAS1R3 as a future target to remediate skeletal muscle insulin resistance in prediabetes and T2D.

## Figures and Tables

**Figure 1 ijms-27-00103-f001:**
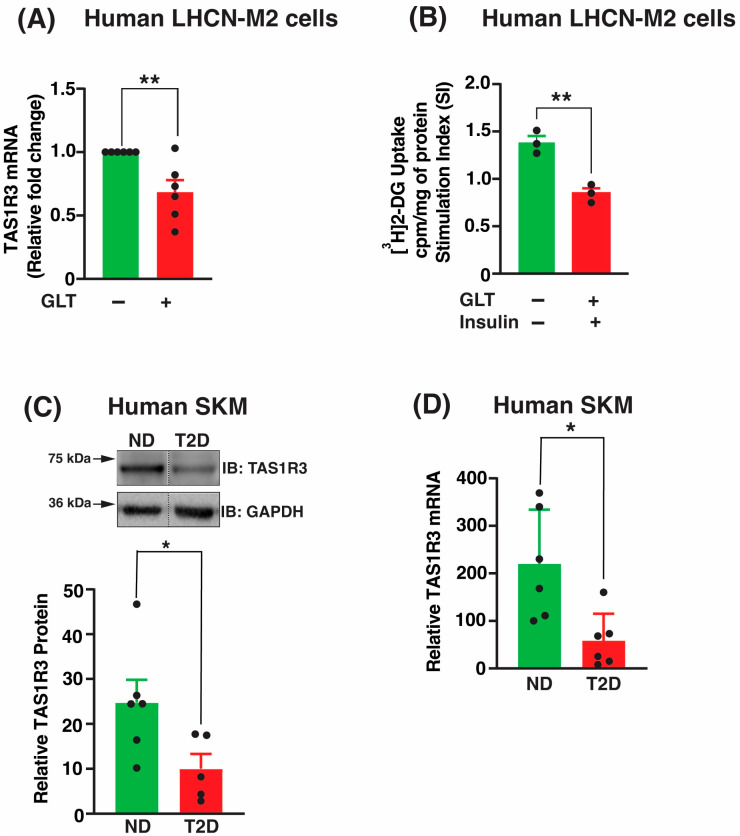
Human T2D skeletal muscle has lower TAS1R3 abundance. (**A**,**B**) LHCN-M2 myotube cells were untreated or exposed to GLT stress for up to 16 h. (**A**) TAS1R3 mRNA was expressed relative to tubulin mRNA. (*n* = 6 independent experiments). (**B**) Glucose ([^3^H]2-DG) uptake ± insulin stimulation was normalized to protein content. Cpm: Counts per minute (*n* = 3 independent experiments). Stimulation index (SI) was calculated as glucose uptake with insulin divided by the glucose uptake at basal conditions. (**C**) Densitometry analysis of TAS1R3 protein in T2D (*n* = 5 donors) versus ND (*n* = 6 donors) skeletal muscle. Top: representative immunoblot (IB). TAS1R3 protein was expressed as relative to GAPDH. Vertical dashed lines indicate the splicing of lanes from within the same gel exposure. (**D**) Quantification of TAS1R3 mRNA expression in human T2D (*n* = 6 donors) versus ND (*n* = 6 donors) skeletal muscle using qPCR. TAS1R3 mRNA was expressed relative to tubulin mRNA. (**A**–**D**). Data are expressed as mean ± SEM. * *p* < 0.05 and ** *p* < 0.01.

**Figure 2 ijms-27-00103-f002:**
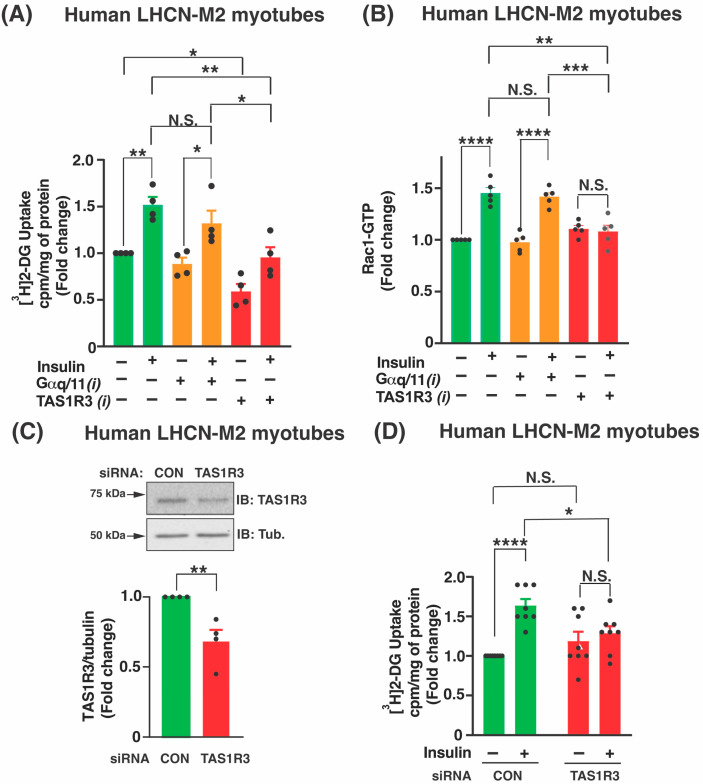
Lactisole inhibits insulin-stimulated glucose uptake and Rac1 activation in human LHCN-M2 myotubes. (**A**,**B**) Human LHCN-M2 myotubes were pre-treated with vehicle (DMSO) or inhibitor for 1 h and then stimulated with ± 100 nM insulin 5 minor for 20 min for Rac1 activation and glucose uptake assay, respectively. TAS1R3(i); lactisole. Gαq/11(i); YM-25490. Assessment of [^3^H]2-DG uptake (**A**) and Rac1 activation (**B**). (**A**) Glucose uptake was normalized to the protein content. Cpm: Counts per minute (*n* = 4 independent experiments). (**B**) Rac1 activation values are relative to baseline Rac1-GTP levels (set to 1.0) detected with unstimulated basal insulin (*n* = 5 independent experiments). (**C**,**D**) Human LHCN-M2 myotubes were pre-treated with siRNA (48 h) and then stimulated with ±100 nM insulin (10 min). CON: control. (**C**) Impact of TAS1R3 siRNA knockdown. Top: Representative immunoblot (IB). Bottom: Densitometry analysis of TAS1R3 protein expression normalized to tubulin (Tub). Control-siRNA values were set to 1.0. *n* = 4 independent experiments. (**D**) Effects of siRNA treatment on glucose uptake with insulin stimulation (*n* = 8 independent experiments). Glucose uptake was normalized to the protein content. Cpm: Counts per minute. Values under basal unstimulated insulin conditions were set to 1.0. (**A**–**D**) Data are expressed as mean ± SEM. * *p* < 0.05, ** *p* < 0.01, *** *p* < 0.001, and **** *p* < 0.0001. N.S: not significant.

**Figure 3 ijms-27-00103-f003:**
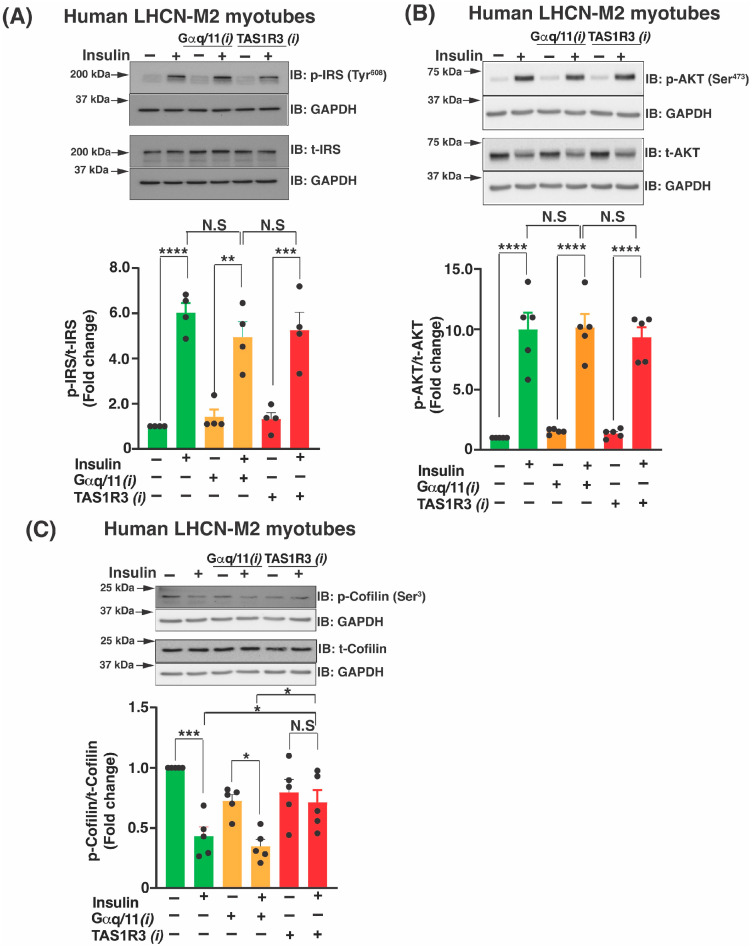
Lactisole fails to inhibit the activation of IRS and AKT but prevents the dephosphorylation of cofilin. (**A**–**C**) Human LHCN-M2 myotubes were pre-treated with vehicle (DMSO), TAS1R3i lactisole, or Gαq/11i YM-25490 for 1 h and were stimulated with ±100 nM insulin (10 min). A representative immunoblot and densitometry analysis is shown for (**A**) phospho-IRS (Tyr^608^) and total IRS (*n* = 4 independent experiments), (**B**) phospho-AKT (Ser^473^) and total AKT (*n* = 5 independent experiments), and (**C**) phospho-cofilin (Ser^3^) and total cofilin (*n* = 5 independent experiments). Data are expressed as mean ± SEM. * *p* < 0.05, ** *p* < 0.01, *** *p* < 0.001, and **** *p* < 0.0001. N.S: not significant.

**Figure 4 ijms-27-00103-f004:**
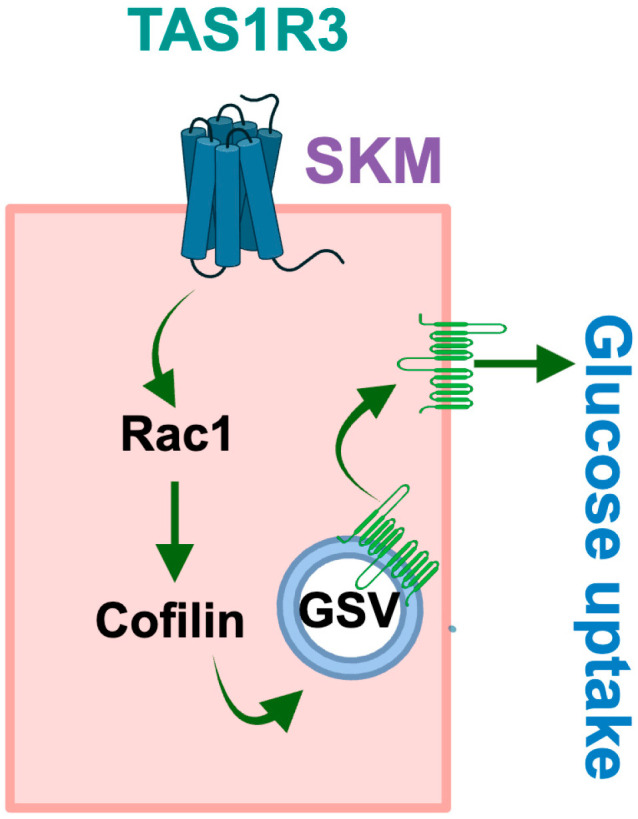
Schematic of TAS1R3 signaling in skeletal muscle. Activation of TAS1R3, a sweet taste receptor expressed in skeletal muscle, triggers a non-canonical signaling cascade that activates Rac1—a small GTPase essential for actin cytoskeletal remodeling. This sustained Rac1 activity prevents the insulin-driven dephosphorylation of Cofilin. Consequently, GLUT4 vesicle trafficking to the plasma membrane is impaired, leading to a marked reduction in glucose uptake.

**Table 1 ijms-27-00103-t001:** Non-diabetic and T2D human skeletal muscle donor profiles.

NDRI#	Age	Race	Gender	BMI	Condition
ND09743	46	Caucasian	M	34.9	ND
ND09744	62	Caucasian	M	28.2	ND
ND09749	49	Caucasian	F	21.6	ND
ND09706	45	Caucasian	M	44.5	T2D
ND09754	67	Caucasian	M	32.3	T2D
ND10105	53	Caucasian	F	28	T2D
ND13347	78	Caucasian	M	26.4	ND
ND13230	68	Caucasian	F	40.2	ND
ND13216	54	Caucasian	M	31.3	ND
ND13214	74	Caucasian	M	32.1	T2D
ND13363	59	Caucasian	M	41.5	T2D
ND13239	58	Caucasian	M	50.3	T2D

ND = Non-diabetic individuals; T2D = Type 2 diabetic individuals; BMI = body mass index; M = bale; F = female; NDRI# = national disease research interchange (NDRI) sample number, provided muscle source as quadriceps or ‘leg muscle’.

## Data Availability

The original contributions presented in this study are included in the article. Further inquiries can be directed to the corresponding authors.

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
