# Peer review of "TAS1R3 Regulates GTPase Signaling in Human Skeletal Muscle Cells for Glucose Uptake"

_ijms, 2025, doi:10.3390/ijms27010103_

Round 1
Reviewer 1 Report
Comments and Suggestions for Authors
The manuscript titled, TAS1R3 regulates GTPase signaling in human skeletal muscle cells for glucose uptake is interesting and scientifically novel. However, this study has the following concerns that need to be addressed to improve its quality.
Major Concerns
- The authors demonstrate that Rac1 activation relies on TAS1R3, but the upstream G-protein mediator is not yet identified. The manuscript discusses Gα14 (Discussion, lines 197–206), but this is only speculative and lacks supporting experimental evidence. Without clarifying the direct coupling mechanism, the proposed model remains incomplete.
- Lactisole acts as a TAS1R3 inhibitor, yet the text does not mention dose-response or control experiments for sweet-taste receptor inhibition. Consequently, conclusions that depend heavily on lactisole should be supported by more robust validation.
- YM-254890 is suitable for blocking Gαq/11; however, the negative outcome might be due to an inadequate inhibitor concentration or exposure duration. Moreover, Gαs is significant in insulin-induced skeletal muscle glucose intolerance, so understanding its role is also crucial.
- The T2D vs. non-diabetic biopsy comparison includes a relatively small sample (n=6 ND, n=6 T2D). Additionally, no HbA1c or duration of diabetes was provided. Thus, observed TAS1R3 differences may be confounded by donor variability.
Minor Concerns
- The representative gel images in Figure 1C include spliced lanes in ND and T2D groups. Running the samples next to each other is very important.
- The biopsy table should include the following parameters:
- HbA1c
- fasting glucose or insulin
- duration of diabetes
- medications (metformin, GLP-1 RAs, insulin)
- physical activity status,
- Medications such as GLP-1 RAs can reduce muscle mass and may influence TAS1R3 levels.
- Including the following information would improve the introduction:
- Other GPCRs that regulate muscle metabolism
- Prior studies on taste receptors in peripheral tissues
- Known links between Rac1 dysregulation and T2D muscle dysfunction
Author Response
Reviewer 1
The manuscript titled, TAS1R3 regulates GTPase signaling in human skeletal muscle cells for glucose uptake is interesting and scientifically novel. However, this study has the following concerns that need to be addressed to improve its quality.
Thank you for your thoughtful review and valuable comments on our manuscript. We appreciate your insights and have revised the manuscript accordingly. Below, we provide a detailed point-by-point response
Major Concern.
1. The authors demonstrate that Rac1 activation relies on TAS1R3, but the upstream G-protein mediator is not yet identified. The manuscript discusses Gα14 (Discussion, lines 197–206), but this is only speculative and lacks supporting experimental evidence. Without clarifying the direct coupling mechanism, the proposed model remains incomplete.
We acknowledge that the precise G-protein coupling mechanism remains unresolved. Our data strongly indicate TAS1R3 regulates Rac1 activation independently of Gαq/11, as YM-254890 treatment did not alter Rac1 activity or glucose uptake. While Gα14 is discussed as a candidate based on literature and FR900359 inhibitor studies, we agree this is speculative. We have revised the Discussion to state Gα14 as a potential candidate out of many based on the prior literature. We then added a statement emphasizing the need for future studies to experimentally validate the specific G-protein involved. We have modified the text “One promising G-protein candidate for future studies based on prior literature is Gα14, based on the finding that FR900359, a selective inhibitor of Gαq/11/14 significantly impairs both GLUT4 translocation and glucose uptake in skeletal muscle cell (PMID: 30936140), which YM-254890, a Gαq/11-specific inhibitor failed to replicate. However, with Gα14 just one example of numerous potential G-protein candidates, future studies will need to focus on identifying the downstream regulators that directly link TAS1R3 activity with Rac1-GTP mediated GLUT4-vesicle translocation for insulin-stimulated glucose uptake in skeletal muscle cells in vitro” (Discussion; lines 200-206).
2. Lactisole acts as a TAS1R3 inhibitor, yet the text does not mention dose-response or control experiments for sweet-taste receptor inhibition. Consequently, conclusions that depend heavily on lactisole should be supported by more robust validation.
We would like to clarify that the concentrations of YM254890 (10 μM) and lactisole (500 μM and 1 mM) employed in our study are consistent with previously published protocols, including our recent paper in Frontiers in Endocrinology (doi: [10.3389/fendo.2025.1695980; Ref 27), where these doses were used without any adverse effects on cell viability or signaling specificity. This reinforces the validity of our experimental design and ensures comparability across studies. Furthermore, to confirm the functional role of TAS1R3 in skeletal muscle cells, we performed siRNA-mediated knockdown of TAS1R3, which significantly attenuated glucose uptake responses. These complementary approaches strengthen the conclusion that TAS1R3 is a key regulator in this context we have modified the text “To determine the requirement for TAS1R3 in skeletal muscle peripheral glucose uptake, we evaluated the impact of TAS1R3 (1 mM Lactisole; TAS1R3(i)) or Gαq/11 (10mM YM-25490; Gαq/11) inhibition against a vehicle control in insulin-stimulated human LHCN-M2 myotubes by measuring 2-deoxyglucose ([3H]2-DG) uptake based on initial dose-response studies for insulin signaling (data not shown) and their prior use in glucose stimulated insulin secretion (GSIS) in pancreatic β-cells (doi: [10.3389/fendo.2025.1695980)” (Results; lines 113-120).
3. YM-254890 is suitable for blocking Gαq/11; however, the negative outcome might be due to an inadequate inhibitor concentration or exposure duration. Moreover, Gαs is significant in insulin-induced skeletal muscle glucose intolerance, so understanding its role is also crucial.
The dose of YM254890 used in our study (10 μM) was chosen based on previously published work in MIN6 β-cells (PMID: 19352508), where it was shown to effectively inhibit Gαq/11 signaling. In that study, the sweet taste receptor expressed in pancreatic β-cells was demonstrated to activate calcium and cyclic AMP signaling pathways, thereby stimulating insulin secretion. Specifically, sucralose-induced activation of TAS1R3 led to elevations in intracellular calcium and cAMP, which were only minimally affected by YM254890 at 10 μM. This observation suggests that Gαq/11 may not serve as the primary mediator of TAS1R3 signaling in β-cells.This supports our rationale for using the same dose, as it reflects physiologically relevant conditions and previously validated experimental parameters in human cell lines or primary islets. Furthermore, our recent paper in Frontiers in Endocrinology (doi.org/10.3389/fendo.2025.1695980) also employed YM254890 at 10 μM and lactisole at 500 μM and 1 mM without any alterations, reinforcing consistency with established protocols.We have clarified that YM-254890 was used at 10 μM for 1 h, consistent with published protocols (PMID: 19352508), for effective Gαq/11 inhibition. To address concerns about Gαs, we have added a statement in the Discussion “One promising G-protein candidate for future studies based on prior literature is Gα14, based on the finding that FR900359, a selective inhibitor of Gαq/11/14 significantly impairs both GLUT4 translocation and glucose uptake in skeletal muscle cells (PMID: 30936140), which YM-254890, a Gαq/11-specific inhibitor failed to replicate. However, with Gα14 just one example of numerous potential G-protein candidates, future studies will need to focus on identifying the downstream regulators that directly link TAS1R3 activity with Rac1-GTP mediated GLUT4-vesicle translocation for insulin-stimulated glucose uptake in skeletal muscle cells in vitro” (Discussion; lines 200-206).
4. The T2D vs. non-diabetic biopsy comparison includes a relatively small sample (n=6 ND, n=6 T2D). Additionally, no HbA1c or duration of diabetes was provided. Thus, observed TAS1R3 differences may be confounded by donor variability. Small biopsy sample size and lack of HbA1c/duration data
We acknowledge the limitation of the relatively small sample size (n = 6 non-diabetic, n = 6 T2D) and the absence of HbA1c values or duration of diabetes for the donor biopsies. These factors could introduce variability and potentially confound TAS1R3 expression differences. However, this was the only information available at the time of analysis, and we have clearly stated this limitation in the manuscript. Despite these constraints, the observed trends provide preliminary insights that warrant further investigation in larger, well-characterized cohorts.
We have also acknowledged this limitation explicitly in the Discussion and emphasized the need for larger cohort studies to confirm these findings. We have updated this in (Discussion; lines 216-224) “We showed that TAS1R3 mRNA and protein levels were depleted in T2D skeletal muscle, which was similarly observed in GLT-exposed human LHCN-M2 myotubes which is used a model for earlier stages of insulin resistance, suggesting that TAS1R3 levels are depleted in insulin resistant muscle. To the best of our knowledge, this is the first study that has associated in clinical context the association of TAS1R3 levels with skeletal muscle insulin resistance. However, we acknowledge that our human biopsy data is limited by small sample pool (n=6 per group) that could be impacted by donor variability based on the available clinical data showing sex, age and body mass index (BMI) differences. Thus, the need for larger cohorts are needed that compose of non-insulin resistant, prediabetes (impaired fasting blood glucose and/or impaired insulin tolerance) and T2D patients to support the decline of TAS1R3 with progression of insulin resistance.”
Minor Concerns
1. The representative gel images in Figure 1C include spliced lanes in ND and T2D groups. Running the samples next to each other is very important.
We clarified that lanes were spliced from the same gel exposure and added this note in the figure legend (Results; line 104-105)
2. The biopsy table should include the following parameters: HbA1c, fasting glucose or insulin duration of diabetes, medications (metformin, GLP-1 RAs, insulin), physical activity status. Medications such as GLP-1 RAs can reduce muscle mass and may influence TAS1R3 levels.
We appreciate the reviewer’s suggestion to include additional clinical and lifestyle parameters such as HbA1c, fasting glucose or insulin, duration of diabetes, medication history (e.g., metformin, GLP-1 receptor agonists, insulin), and physical activity status in the biopsy table. We agree that these factors could influence skeletal muscle physiology and TAS1R3 expression, particularly since GLP-1 receptor agonists have been reported to reduce muscle mass.
However, the donor biopsy samples were obtained through the National Disease Research Interchange (NDRI), and the only information provided at the time of analysis included age, race, gender, BMI, and diabetic status (non-diabetic vs. T2D). Unfortunately, HbA1c values, duration of diabetes, medication history, and physical activity data were not available. We have clearly acknowledged this limitation in the manuscript and noted that the relatively small sample size (n = 6 non-diabetic, n = 6 T2D) and absence of these clinical parameters could introduce variability and potentially confound TAS1R3 expression differences.
Future studies will aim to incorporate these critical clinical and lifestyle variables to better delineate their impact on TAS1R3 signaling in skeletal muscle.
3. Including the following information would improve the introduction: Other GPCRs that regulate muscle metabolism. Prior studies on taste receptors in peripheral tissues. Known links between Rac1 dysregulation and T2D muscle dysfunction
We appreciate the reviewer’s recommendation to strengthen the introduction by including (i) other GPCRs that regulate muscle metabolism, (ii) prior studies on taste receptors in peripheral tissues, and (iii) known links between Rac1 dysregulation and T2D muscle dysfunction. We agree that these points provide valuable context and have revised in Introduction lines 60–75 accordingly.
The revised section now highlights that, in addition to TAS1R3, other GPCRs such as βâ‚‚-adrenergic receptors play critical roles in skeletal muscle hypertrophy and atrophy, as well as in adaptive responses to anabolic and catabolic conditions. We have also cited representative studies on taste receptors in non-gustatory tissues (e.g., pancreatic β-cells, gastrointestinal tract, musculoskeletal system) (Introduction lines 60–75) accordingly and briefly noted the importance of Rac1 in insulin-stimulated glucose uptake and its dysregulation in T2D muscle, which is relevant to the signaling pathways discussed in this study. These additions provide a broader context for TAS1R3’s potential role in skeletal muscle metabolism while maintaining the manuscript’s focus (Introduction; lines 79-82).
Reviewer 2 Report
Comments and Suggestions for Authors
This study by Hoolachan et al shows that the taste receptor TAS1R3 expression is reduced in skeletal muscle of people with type 2 diabetes, and its expression is essential for insulin-stimulated glucose uptake. Using human LHCN-M2 myotubes, the authors show that TAS1R3 acts through a non-canonical Rac1–cofilin pathway to trigger GLUT4 translocation to the plasma membrane. This pathway is independent of IRS1–AKT and Gαq/11 signaling, making it a potential novel therapeutic target for improving insulin sensitivity.
Minor comments:
- Are there known mutations or SNPs in the TAS1R3 gene associated with increased risk of T2D? It would be helpful for a novel therapeutic perspective to include this background if data is available.
- Although TAS1R3 signaling pathway is explained in the introduction, it would benefit readers to know more about the endogenous ligands which activate TAS1R3 and whether levels of TAS1R3 ligands are altered in obesity/ T2D pathogenesis.
- Besides its effect on GLUT4 trafficking, is TAS1R3 signaling known to affect GLUT-4 gene expression?
- TAS1R3 control of basal glucose uptake is interesting. Can the authors speculate on whether TAS1R3 expression is reduced in people with impaired fasting blood glucose without overt diabetes?
Author Response
Reviewer # 2
This study by Hoolachan et al shows that the taste receptor TAS1R3 expression is reduced in skeletal muscle of people with type 2 diabetes, and its expression is essential for insulin-stimulated glucose uptake. Using human LHCN-M2 myotubes, the authors show that TAS1R3 acts through a non-canonical Rac1–cofilin pathway to trigger GLUT4 translocation to the plasma membrane. This pathway is independent of IRS1–AKT and Gαq/11 signaling, making it a potential novel therapeutic target for improving insulin sensitivity.
Thank you for your thoughtful review and valuable comments on our manuscript. We appreciate your insights and have revised the manuscript accordingly. Below, we provide a detailed point-by-point response.
Minor comments:
1.Are there known mutations or SNPs in the TAS1R3 gene associated with increased risk of T2D? It would be helpful for a novel therapeutic perspective to include this background if data is available.
We searched the literature and found no direct evidence linking TAS1R3 SNPs to T2D risk. However, TAS1R3 polymorphisms (e.g., rs307355, rs35744813) have been associated with variations in sweet taste perception and dietary sugar intake, which indirectly influence metabolic health. We have added this information to the Introduction and noted that while TAS1R3 variants may affect nutrient sensing, their direct role in T2D pathogenesis remains to be established. We have now modified the text “Although prior literature have no direct evidence linking TAS1R3 single nucleotide polymorphisms with type 2 diabetes (T2D) and obesity risks, studies on gastric tissue and pancreatic-beta cells have shown diminished TAS1R3 expression in obese and T2D patients, suggesting that TAS1R3 activity may be altered in obesity/T2D pathogenesis” (Introduction; lines 69-72).
2. Although TAS1R3 signaling pathway is explained in the introduction, it would benefit readers to know more about the endogenous ligands which activate TAS1R3 and whether levels of TAS1R3 ligands are altered in obesity/ T2D pathogenesis.
We have expanded the Introduction to include details on endogenous ligands: TAS1R3 responds to glucose, sucrose, and certain amino acids. While circulating glucose is elevated in T2D, whether TAS1R3 ligands are dysregulated in skeletal muscle microenvironment is unknown. We have acknowledged this gap and proposed future studies to investigate ligand availability in obesity/T2D. We have now modified the introduction per the reviere request like this “Normally, the nutrient sensing role of TAS1R3 is dependent upon its dimerization, with TAS1R3/TAS1R3 homodimers and TAS1R2/TAS1R3 heterodimers serving as glucose sensors and TAS1R1/TAS1R3 heterodimers serving as an amino acid sensor. Importantly, downregulation of TAS1R3 levels coincide with abberant whole body glucose homeostasis as observed with the impaired glucose tolerance in global TAS1R3 knockout mice. Although prior literature have no direct evidence linking TAS1R3 single nucleotide polymorphisms with T2D and obesity risks, studies on gastric tissue and pancreatic-beta cells have shown diminished TAS1R3 expression in obese and T2D patients, suggesting that TAS1R3 activity may be altereted in obesity/T2D pathogenesis. Given that a number of conserved molecular pathways play a role in both β-cell GSIS and skeletal muscle insulin-stimulated glucose uptake (35-40), it still remains to be determined if skeletal muscle TAS1R3 expression is altered during T2D pathogenesis and whether this is attributed to a potential novel role of TAS1R3 in insulin-stimulated glucose uptake glucose uptake” (Introduction; lines 65-75).
3. Besides its effect on GLUT4 trafficking, is TAS1R3 signaling known to affect GLUT-4 gene expression?
To our knowledge, TAS1R3 signaling does not directly regulate GLUT4transcription. Both our data and prior studies indicate that its role is primarily post-translational, facilitating GLUT4vesicle translocation through Rac1-mediated actin remodeling. We have clarified this point in the Discussion. Importantly, these experiments involved short-term exposure to TAS1R3 agonists; therefore, we do not anticipate any changes in GLUT4 expression levels under these conditions (Discussion; lines 196-199).
4. TAS1R3 control of basal glucose uptake is interesting. Can the authors speculate on whether TAS1R3 expression is reduced in people with impaired fasting blood glucose without overt diabetes?
We agree this is an important question. Although our study focused on T2D, we state that the glucolipotoxicity-exposed LHCN-M2 myotubes serves as a model for earlier stages of insulin resistance, which TAS1R3 downregulation was observed. We do highlight as a limitation the need for prediabetes cohorts who may be characterized by impaired fasting blood glucose and/or impaired insulin sensitivity to strengthen the association between TAS1R3 levels and insulin sensitivity. We have updated this in (Discussion; lines 216-224) “We showed that TAS1R3 mRNA and protein levels were depleted in T2D skeletal muscle, which was similarly observed in GLT-exposed human LHCN-M2 myotubes which is used a model for earlier stages of insulin resistance, suggesting that TAS1R3 levels are depleted in insulin resistant muscle. To the best of our knowledge, this is the first study that has associated in clinical context the association of TAS1R3 levels with skeletal muscle insulin resistance. However, we acknowledge that our human biopsy data is limited by small sample pool (n=6 per group) that could be impacted by donor variability based on the available clinical data showing sex, age and body mass index (BMI) differences. Thus, the need for larger cohorts are needed that compose of non-insulin resistant, prediabetes (impaired fasting blood glucose and/or impaired insulin tolerance) and T2D patients to support the decline of TAS1R3 with progression of insulin resistance”.
Round 2
Reviewer 1 Report
Comments and Suggestions for Authors
The authors addressed all of my concerns with the previous version of the manuscript. Therefore, I recommend the acceptance of this revised version.